# JAK2 Unmutated Polycythaemia—Real-World Data of 10 Years from a Tertiary Reference Hospital

**DOI:** 10.3390/jcm11123393

**Published:** 2022-06-13

**Authors:** Katarzyna Aleksandra Jalowiec, Kristina Vrotniakaite-Bajerciene, Jakub Jalowiec, Noel Frey, Annina Capraru, Tatiana Wojtovicova, Raphael Joncourt, Anne Angelillo-Scherrer, Andre Tichelli, Naomi Azur Porret, Alicia Rovó

**Affiliations:** 1Department of Haematology and Central Haematology Laboratory, University Hospital/Inselspital Bern, University of Bern, 3010 Bern, Switzerland; kristina.vrotniakaite-bajerciene@insel.ch (K.V.-B.); annina.capraru@usz.ch (A.C.); tatiana.wojtovicova@insel.ch (T.W.); raphael.joncourt@insel.ch (R.J.); anne.angelillo-scherrer@insel.ch (A.A.-S.); naomiazur.porret@insel.ch (N.A.P.); alicia.rovo@insel.ch (A.R.); 2Independent Researcher, 02-776 Warsaw, Poland; kuba.jalowiec@protonmail.com; 3IDSC Insel Data Science Center, University Hospital/Inselspital Bern, University of Bern, 3010 Bern, Switzerland; noel.frey@insel.ch; 4Haematology, University Hospital of Basel, 4031 Basel, Switzerland; andre.tichelli@gmail.com

**Keywords:** polycythaemia, erythrocytosis, JAK2 unmutated, underlying causes, thrombotic risk, work-up

## Abstract

(1) Background: Polycythaemia is defined by an increase in haemoglobin (Hb) concentration, haematocrit (Hct) or red blood cell (RBC) count above the reference range adjusted to age, sex and living altitude. JAK2 unmutated polycythaemia is frequent but under-investigated in original publications. In this retrospective cohort study, we investigated the clinical and laboratory data, underlying causes, management and outcomes of JAK2 unmutated polycythaemia patients. (2) Methods: The hospital database was searched to identify JAK2 unmutated patients fulfilling WHO 2016 Hb/Hct criteria for PV (Hb >16.5 g/dL in men and >16 g/dL in women, or Hct > 49% in men and >48% in women, or RBC mass > 25% above mean normal predicted value) between 2008 and 2019. Clinical and laboratory data were collected and analysed. (3) Results: From 727,731 screened patients, 294 (0.04%) were included, the median follow-up time was 47 months. Epo and P50 showed no clear pattern in differentiating causes of polycythaemia. In 30%, the cause remained idiopathic, despite extensive work-up. Sleep apnoea was the primary cause, also in patients under 30. Around 20% had received treatment at any time, half of whom had ongoing treatment at the end of follow-up. During follow-up, 17.2% developed a thromboembolic event, of which 8.5% were venous and 8.8% arterial. The mortality was around 3%. (4) Conclusions: Testing for Epo and P50 did not significantly facilitate identification of underlying causes. The frequency of sleep apnoea stresses the need to investigate this condition. Idiopathic forms are common. A diagnostic flowchart based on our data is proposed here. NGS testing should be considered in young patients with persisting polycythaemia, irrespective of Epo and P50 levels.

## 1. Introduction

Polycythaemia is defined by an increase in haemoglobin (Hb) concentration, haematocrit (Hct) or red blood cell (RBC) count above the reference range adjusted to age, sex and living altitude. JAK2 unmutated polycythaemia is frequent in clinical practice and is usually classified into polycythaemia vera (PV), secondary and hereditary polycythaemia. Nowadays, with the discovery of JAK2 driver mutations, PV no longer represents a diagnostic difficulty. A more serious challenge is JAK2 unmutated polycythaemia, which encompasses a heterogenous group of entities.

According to the revised World Health Organization (WHO) classification of myeloid neoplasms from 2016, there are clear diagnostic criteria for PV, including haemoglobin (Hb) > 16.5 g/dL in men and Hb > 16 g/dL in women, or haematocrit (Hct) > 49% in men and Hct > 48% in women, or red blood cell (RBC) mass > 25% above mean normal predicted value [1]. For the diagnosis of JAK2 unmutated polycythaemia, usually the PV Hb/Hct criteria are applied. Patients who present with polycythaemia and do not have PV driver mutations, namely JAK2 V617F and exon 12, should be investigated for secondary causes of polycythaemia. Data regarding the epidemiology of secondary polycythaemia is sparse.

Hereditary causes of JAK2 unmutated polycythaemia should be investigated, especially in cases of long-term polycythaemia, in young patients and if family history is positive. According to the literature [2,3], testing for erythropoietin (Epo) and partial pressure of oxygen at which 50% of haemoglobin is oxygen-saturated (P50) may be helpful here. The search for underlying hereditary causes, according to Epo and P50 levels, is suggested [2] (Figure 1).

In clinical practice, therapy guidelines for PV are frequently applied for the management of JAK2 unmutated polycythaemia, which may result in overtreatment of these benign conditions.

There is a myriad of publications outlining the diagnostic work-up of JAK2 unmutated polycythaemia. However, since the discovery of JAK2 V617F in 2005, the vast majority of them are reviews based on expert suggestions, not on original research work. There are only singular studies that systematically analysed causes, diagnostic steps and outcomes, including thrombotic events in patients with JAK2 unmutated polycythaemia [4,5].

Therefore, in this retrospective cohort study, we aimed to investigate the clinical and laboratory data of underlying causes, as well as the real-world management and outcomes, of patients with JAK2 unmutated polycythaemia at a Swiss tertiary reference centre.

## 2. Methods

### 2.1. Cohort Definition

For this study, we assumed that in every patient in whom no clinical underlying cause was found to explain polycythaemia, JAK2 V617F mutation should have been investigated. As a result, we focused on those patients whose laboratory confirmed polycythaemia and where JAK2 V617F mutation was tested, the result was available, and it was negative. Using this stepwise identification process (Figure 2), patients with polycythaemia, with a negative JAK2 V617F result, constituted our cohort. The laboratory and potential clinical causes for polycythaemia are reported.

The hospital database (Microsoft SQL Server Management Studio) was searched to identify polycythaemia patients in the study period between 1 October 2008 and 31 July 2019. This was performed using different strategies: for inpatients, we used the ICD-10 coding system (D45 for polycythaemia vera or D75.1 for secondary polycythaemia); for outpatients, key words were used (polycythaemia, erythrocytosis, polycythaemia vera, PV). All patients ≥ 15 years of age were included in the search. All identified patients were individually evaluated to confirm if they met the Hb/Hct WHO 2016 criteria for PV.

### 2.2. Laboratory and Clinical Data Source

All laboratory data was filtered and collected automatically using our hospital’s laboratory system (Labormanagement software OPUS::L, Dedalus Labor GmbH-OSM Gruppe, Essen, Germany). If more than one measurement was available for full blood count (FBC), Epo or P50, the nearest measurement to identification of JAK2 V617f mutation was chosen for analysis. Laboratory results not automatically retrievable, such as molecular reports, were manually acquired from the clinical records. The clinical data of patients was manually collected by a clinical record review. Alive/dead status and date of death, if applicable, was determined from the hospital or state insurance database.

### 2.3. Definitions

The follow-up period was defined as the time between first documented polycythaemia, according to WHO 2016 criteria, and the last available Hb/Hct values. The data collection cut-off was 31 July 2019. The availability of a JAK2 V617F result was considered crucial to definitively rule out the diagnosis of PV. Patients were considered lost to follow-up if no second Hb/Hct measurement was performed. To evaluate patients’ outcomes, the occurrence of new venous or arterial thrombosis and death were included. Data regarding thromboembolic events were retrieved from clinical records and only documented (not suspected) events were taken into account. Young patients were defined as being under 30 years of age.

### 2.4. Causes of Polycythaemia

The causes of polycythaemia were divided into 5 groups: (1) relative (spurious); (2) reactive, oxygen-driven, consisting of all cases where low cellular oxygen seems to play a role; (3) reactive Epo or hormonal-driven, with cases in which an endogenous or exogenous substance seems to play a role; (4) congenital, including those with proven molecular abnormality, congenital hemoglobinopathy and also Down’s Syndrome, known to be associated with polycythaemia; (5) patients with an undetermined (idiopathic) cause (Table 1). If we identified several causes that could explain polycythaemia, the most likely was considered for characterisation. Doubtful cases were evaluated via second opinion by an experienced haematologist with expertise in myeloproliferative neoplasms (MPN) and RBC disorders.

We classified cases as relative polycythaemia only if normal RBC mass was confirmed on nuclear measurement of the ratio between erythrocyte volume and plasma volume (EV/PV). RCM and plasma volume were obtained using chromium-labelled RBC and albumin, respectively. Dehydration, Gaisböck syndrome and capillary leak syndrome were considered as causes only if confirmed by a clinician. Sleep apnoea was classified as a cause only if it was confirmed by a polysomnography. Altitude was classified as a cause only if the patient lived above 1500 metres over the sea level. Smoking was classified as a cause only if the patient was an active smoker at the time of diagnosis of polycythaemia, irrespective of pack-years smoked. Respiratory disease, such as chronic obstructive pulmonary disease (COPD) and pneumopathy, were considered as causes only if a diagnosis was confirmed by a pneumonologist. Cardiac diseases were considered as causes only when there was a right–left shunt resulting in a cyanotic heart disease confirmed by a cardiologist. Increased Epo was considered as a cause if it was above the reference range used at our centre (4.30–29.0 mU/mL). Noncancer kidney diseases were only classified as causes when there seemed to be a clear renal arterial problem, for instance one small kidney or atherosclerotic stenosis vessels, or in cases of hydronephrosis or real cystic diseases. Androgen therapy was considered as a cause only if a patient was being treated with androgens when polycythaemia was diagnosed. As far as neoplasia related to polycythaemia is concerned, we only included cancers which have been published to be related with polycythaemia, and therefore the following cancers were included in this group: renal [6,7,8]; adrenal [9,10]; genitourinary [11,12]; respiratory [13]; cerebral tumours [14]; hepatocellular cancer [15]. Patients with any other type of cancer were included in the idiopathic group. Haemoglobinopathy was considered as a cause only if a laboratory report confirming the diagnosis was available. High affinity Hb was considered a cause only if P50 was <24 mmHg [8].

### 2.5. Statistical Analysis

The data collection and analysis were conducted in Excel Statistics 2019, IBM SPSS Statistics for Windows (version 27), and R-statistics (version 4.1.2). A non-normal distribution of all variables was assumed, and median and range values were reported. The Kruskal–Wallis test was used to compare medians of three or more groups. The frequencies of categorical variables were compared using Pearson’s χ^2^ test. A value of *p* < 0.05 was considered significant in both tests. For the follow-up analysis, 13 patients who did not have a repeated Hb/Hct measurement were excluded. The missing laboratory data were excluded from analysis.

## 3. Results

### 3.1. Study Population

The search was conducted across available medical reports of 727,731 patients, 239,516 inpatients and 488,215 outpatients with any record fitting our criteria.

We identified 4391 patients whose medical records reported polycythaemia (or a similar keyword). Of these, 282 occurrences were duplicates (both inpatient and outpatient) and therefore their records were merged. The remaining 4109 patients that represent 0.56% of all screening patients were examined for Hb/Hct WHO 2016 criteria for PV. In this process, we confirmed 1483 patients (0.2% of all screened patients) fulfilling the Hb/Hct criteria. From 1483 patients, 391 were tested for JAK2 V617 mutation with 297 (76%) of them being JAK2 V617F negative and 94 (24%) positive. Thus, these 94 JAK2 V617F positive MPN patients were excluded from our analysis. Additionally, three patients were excluded due to the presence of another clonal mutation (CALR mutation (N = 2); JAK2 exon 12 (N = 1)). In total, 294 patients (0.04% of all screened patients), were classified as having JAK2 unmutated polycythaemia, and they represent the final cohort of this study. Their laboratory and clinical data, as well as the management and outcome, were evaluated and reported here. The flowchart of patient selection is presented in Figure 2.

### 3.2. Causes of JAK2 Unmutated Polycythaemia

In 205 of 294 patients with JAK2 unmutated polycythaemia (70% of our study cohort), an underlying physiopathological mechanism to explain polycythaemia was found. Among those, 14 (5%) patients were diagnosed with relative (spurious), 139 (47%) with reactive oxygen-driven, 40 (14%) with reactive Epo- or hormone-driven and 12 (4%) with congenital polycythaemia. In 89 (30%) patients, the cause was idiopathic. Causes were subclassified, as presented in Table 1 and Table 2 and Figure 3.

The causes of JAK2 unmutated polycythaemia according to the frequency of their occurrence are also presented in Figure 3. Sleep apnoea was the most frequent cause (55, 19%), followed by nicotine use (51, 17%), increase in HbCO > 5% (14, 5%), respiratory diseases (13, 4%), noncancer kidney disease (12, 4%), androgen therapy (9, 3%), relative erythrocytosis with normal RBC mass (9, 3%), cancer (renal, adrenal, seminoma, lung, brain) (8, 3%), renal transplantation (7, 2%) and other causes (27, 9%). Congenital erythrocytosis was identified in 12/294 patients (4%).

Among young patients (56, 19% of all), determined cause (31, 55% of young patients) was more frequent than unexplained polycythaemia (25, 45% of young patients). Moreover, in this group, the most frequently determined causes of polycythaemia were sleep apnoea (6, 11% of young patients) and smoking (6, 11% of young patients). In 17 young patients with persistent polycythaemia at the end of follow-up, the cause was unexplained. We are aiming to investigate this population for congenital causes of polycythaemia as well as myeloproliferative and myelodysplastic diseases.

### 3.3. Demographics and Laboratory Values

There was a great over-representation of males (242, 82%) in the cohort, but the ratio of males to females in each of the five subgroups (Table 1) was not significantly different s (Pearson’s χ^2^ all *p*-values > 0.10).

In contrast, the median age at diagnosis of polycythaemia was statistically different according to the pathophysiological underlying attributed mechanisms of polycythaemia (Kruskal–Wallis Test, *p* = 0.007). Specifically, patients have higher median age in all groups other than congenital polycythaemia, and idiopathic polycythaemia (Table 1). The analysis of laboratory values did not show significant differences between the five presented groups (Kruskal–Wallis Test for haemoglobin *p* = 0.238; haematocrit *p* = 0.521 and RBC *p* = 0.322).

The median follow-up time was 47 months (range 0.4–258), though in 13/294 (4%) patients, follow-up was not available. In the last available blood results, the majority of patients 159/281 (57%) still fulfilled Hb/Hct WHO 2016 criteria for PV, indicating persistent polycythaemia. In 135/281 (48%) patients, polycythaemia resolved during the follow-up period: in 13/135 with treatment and in 118/135 without any intervention (4 patients had unknown status). At the time of the last follow-up, 53 patients had persistent, idiopathic polycythaemia (males N = 48, females N = 5) and 17 of these were younger than 30 years of age and were exclusively male. There were no clinical or laboratory data on thrombophilia available.

For the diagnostic work-up, in 153/294 (52%) patients, the Epo level was tested, in 240/294 (82%), the P50 was tested, 54/294 (18%) were evaluated with a bone marrow examination, 62/294 (21%) with a stem cell culture, 48/294 (16%) with an EV/PV and in 98/294 (33%), for JAK2 exon 12 mutation. Epo was suppressed in 15/153 of Epo-tested patients (10%). Exactly 113 were tested for both Epo and P50. Epo level was normal or increased and P50 was <24 mmHg in 20/113 of those tested (18%), whereas the Epo level was normal or increased and P50 was ≥24 mmH in 93/113 (82%). Of the 196 patients in whom JAK2 exon 12 was not tested, PV was excluded with bone marrow investigation in 22, whereas 95 had persistent polycythaemia at the end of follow-up, but only 3 had suppressed Epo. In the 3/4 of patients with suppressed Epo and idiopathic cause, JAK2 exon 12 was tested, and it was negative. Specific conditions according to Epo and P50 levels are listed in Figure 4.

### 3.4. Investigations of JAK2 Unmuteded Polycythemia

In 227 cases (77%), a haematologist was involved in the diagnostic work-up and/or management of polycythaemia independent of the underlying cause. The only exception to this was observed in patients in which reactive Epo or hormone-driven erythrocytosis was diagnosed, where the majority (N = 21, 53%) were not seen by a haematologist (Pearson’s χ^2^, *p* < 0.001). The median time between the JAK2 V617F negative report and involvement of a haematologist was 25 days (range 0 to +7306), and 29 patients were seen by a haematologist before the diagnosis of JAK2 unmutated polycythaemia due to other issues.

### 3.5. Management of JAK2 Unmutated Polycythaemia

We evaluated the type of therapy that patients with JAK2 unmutated polycythaemia received, and the results are presented in Table 2. We found that a minority (61, 20.7%) had at some time had a treatment of polycythaemia. A total of 53 patients had phlebotomy (18%), 17 received aspirin (ASS) (6%) and 2 received cytoreductive treatment. The type of treatment was not statistically different between the analysed subgroups 1–4 (Pearson’s χ^2^, *p* = 0.149; with exclusion of the idiopathic group (*p* = 0.160). Among these patients, some received more than one type of treatment. The majority were treated with phlebotomy only, followed by phlebotomy with ASS and ASS only. Few patients were treated with combination of three types of therapy. At the end of follow-up, 22/61 (36% of ever treated) had ongoing therapy (Appendix A). The interpretation of treatment with ASS is more vague, since its indication might not necessarily be related to polycythaemia. We also analysed the side effects of the implemented treatment strategy. In 10/61 (16%) treated patients, treatment side effects were reported, including anaemia and/or iron deficiency in 8/10 (80%), and nausea, dizziness or syncope in 2/10 (20%). At the end of follow-up, 22/61 (36%) patients were still undergoing therapy.

### 3.6. Thromboembolic Events in JAK2 Unmutated Patients

During follow-up, a number of thromboembolic events were observed. In total, 33 venous in 25/294 (8.5%) patients and 32 arterial in 26/294 (8.8%) were identified (Appendix A). No patients had both. Patients suffered equally frequently from venous and arterial thromboembolisms (Pearson’s χ^2^
*p* = 0.14). The most frequent venous thromboembolic event was pulmonary embolisms (16/33 events, 49%) and deep venous thrombosis (10, 30%), followed by splanchnic (2, 6%), catheter associated (2/33 events, 6%), ophthalmic (1, 3%), cerebral (1, 3%) and cardiac with one right atrial thrombus (3%). The most frequent arterial thrombotic event was stroke (16/32 events; 50%), followed by myocardial infarction (11, 34%) and other arterial peripheral thromboembolisms (5, 16%). Mostly, patients suffered from a single event (19/25 for venous thromboembolism and 22/26 for arterial thromboembolism). However, some suffered from two (5/25 for venous and 2/25 for arterial) or more events (four venous thromboses in 1/25 and three arterial thromboses in 2/25 of patients). Among eight patients with cancer, two had one venous thromboembolic event. There was no significant gender difference in the prevalence of venous and arterial thromboembolism: venous thromboembolic events in males (21/25, 84%) and females (4/25, 16%) (Pearson’s χ^2^, *p* = 0.81); arterial thromboembolism in males (20/26, 77%) and females (N = 6, 23%) (Chi-square according Pearson’s *p* = 0.45). Furthermore, we analysed the occurrence of thromboembolism according to the age of patients by dividing them into three age groups: 0–30 (N = 54), 31–50 (N = 105) and >50 years old (N = 122). There were no significant differences between the three age groups for both venous and arterial thromboembolism: venous thromboembolism: age 0–30: 3/54 (6%); age 31–50: 10/105 (10%); age >51: 12/122 (10%; Kruskal–Wallis test, *p* = 0.63); arterial thromboembolism: age 0–30: 3/54 (6%); age 31–50: 8/105 (8%); age >51: 15/122 (12%; Kruskal–Wallis test, *p* = 0.28). 

Venous thromboembolism occurred before the diagnosis of polycythaemia in some patients (N = 52, 18% of all and N = 41, 14% of all, respectively; Pearson’s χ^2^
*p* = 0.26).

Further, there were no significant differences in occurrence of venous thromboembolisms when considering the underlying physiopathological mechanisms between the 1–5 subgroups (Pearson’s χ^2^, *p* = 0.92). Likewise, there was no significant difference in prevalence of arterial thromboembolism between the five subgroups (Pearson’s χ^2^, *p* = 0.68). In the idiopathic group, there were 7/33 (21%) venous and 7/32 (27%) arterial thrombotic events. We have not observed any thromboembolism in three patients with mutations causing congenital polycythaemia identified by our in-house Next Generation Sequencing (NGS) panel for polycythaemia [16].

### 3.7. Mortality of JAK2 Unmutated Patients

For the survival analysis, 10 patients (3.4%) with unknown alive/dead status were excluded. At the end of follow-up, the majority were alive (276/284, 97%). Among the eight who died, the majority were in the reactive oxygen-driven (6/135, 4%) and some in the Epo/hormone-driven erythrocytosis (2/39, 5%) group; however, the exact causes of their deaths are unknown.

In our cohort of patients, we have not observed any MPN during the follow-up.

## 4. Discussion

This study, as one of the few since the introduction of JAK2 V617F testing in 2005 [17,18], systematically evaluates the prevalence, causes, management and outcome of a large group of patients with JAK2 unmutated polycythaemia that were seen in a tertiary reference centre.

These real-world data, extracted from clinical practice patients’ files, showed a prevalence of polycythaemia fulfilling Hb/Hct criteria for PV in 0.2% of all screened patients. Furthermore, the prevalence of JAK2 unmutated polycythaemia was higher (N = 294, 0.04% of all screened patients, 40 per 100 000) than the prevalence of JAK2 mutated polycythaemia (N = 94 for V617F and N = 1 for JAK2 exon 12, 0.013% and 0.0014% of all screened patients, respectively, 12 per 100,000). The prevalence of PV in our cohort is lower than previously reported by others, at 44–57 per 100,000 [19].

We confirmed the diagnostic challenge of JAK2 unmutated polycythaemia. Indeed, though a cause was identified in nearly two thirds of patients, in the remaining third, despite extensive work-up, the cause remained unexplained. Interestingly, sleep apnoea was the single most frequent identifiable cause in any age group. The likelihood of sleep apnoea as a cause of reactive polycythaemia is probably underestimated, particularly in younger patients, and our data show the need to emphasise it.

We have also observed that no systematic sequential diagnostic work-up of JAK2 unmutated polycythaemia was followed. It seems sensible to recommend that once PV, as well as obvious secondary causes of polycythaemia, are excluded (i.e., COPD; cyanotic congenital heart disease), the next step should be to investigate for sleep apnoea, irrespective of patients’ age, including a referral for respiratory evaluation and measurement of carboxyhaemoglobin (COHb).

For the interpretation of results, we applied diagnostic algorithms proposed by others [2,3] to our cohort, comprising of sequential testing of Epo and P50 to evaluate for potential congenital causes of JAK2 unmutated polycythaemia. We observed a variety of different entities intermingled in the different combinations (Figure 4). Testing for Epo and P50 did not allow a clear triage for identification of congenital causes.

We have noticed that JAK2 exon 12 mutation was investigated in only one third of our cohort. Indeed, JAK2 exon 12 should be explored systematically, particularly if Epo is suppressed [20]. A possibility to clarify the low rate of JAK2 exon 12 determinations in this cohort is that some patients were seen in the haematology clinic before JAK2 exon 12 mutation was described in the literature in 2007 [20], and its testing was available in clinical practice only after these consultations.

Imaging can make a clear contribution while investigating underlying causes of polycythaemia, such as benign kidney pathology (renal artery stenosis; renal cysts; postrenal transplantation), adrenal disorders (Cushing’s syndrome), as well as Epo-producing tumours (renal tumour; hepatocellular carcinoma; uterine leiomyomas; cerebellar hemangioblastomas, meningioma; pheochromocytoma, paraglioma). However, the most appropriate time to suggest imaging in the work-up algorithm is difficult to define. Probably, clinical judgment is the best strategy for not missing a relevant diagnosis. The measurement of the RBC mass by EV/PV examination, either confirming or ruling out absolute erythrocytosis, is a diagnostic step that in some patients will be essential for a better diagnosis. In recent years, the inclusion of NGS has made an enormous contribution in the diagnosis of congenital forms of polycythaemia [21,22]. This testing should be mainly considered in young patients with persisting polycythaemia, irrespectively of Epo and P50 levels. Offering a broad NGS panel covering mutations that may explain polycythaemia is a valuable option. Our NGS panel was first introduced in 2018 in our centre, which may explain the low rate of use of the method in patients with idiopathic causes. Based on our results, a diagnostic algorithm to investigate JAK2 unmutated polycythaemia is proposed in Figure 5. The list of 13 genes included in the panel, as described elsewhere [16], can be found in the footnote to Figure 5. As a result of limited diagnostic potential using high-performance liquid chromatography (HPLC) [23], as it may not detect high oxygen affinity hemoglobinopathies, our NGS panel includes mutations in α- or β-globin genes (HBB, HBA1, HBA2) and the use of HPLC is not part of our algorithm. However, it does not include mutations in the PIEZO1 gene. Mutations in this gene have been known to cause erythrocyte volume disorders, especially hereditary xerocytosis (HX)/dehydrated stomatocytosis (DHS), which are believed to cause mild-to-moderate haemolytic anaemia. Though, a study from 2019 with a large series of patients with HX and PIEZO1 mutations reported that 68% did not show any sign of anaemia, and seven patients had polycythaemia [24]. Moreover, a study from 2021 found HX, with well-compensated haemolysis, in 4% of investigated idiopathic erythrocytosis [25]. The authors concluded about the need to include the search for a mutation in the PIEZO1 gene in children and young adults with idiopathic erythrocytosis and more generally when erythrocytosis appears before the age of 50 years, especially when associated with iron overload, splenomegaly, elevated mean corpuscular haemoglobin concentration, increased reticulocytes, haemolysis or decreased P50 (after excluding a variant of Hb with high oxygen affinity). Only recently, our laboratory included investigations on PIEZO1 gene mutations.

We found that in almost half of patients, polycythaemia was no longer present at the end of follow-up, suggesting that in many patients there may be transient triggers, or that appropriate interventions such as smoking cessation, use of devices to improve night-time breathing, weight loss and/or discontinuing androgen treatments might have contributed to treat polycythaemia.

In this cohort, a haematologist was involved in the diagnostic work-up and/or management of polycythaemia independent to the underlying cause in 77% of patients. This high percentage may be related to the characteristics of patients themselves, in whom JAK2 testing was considered necessary, but it may also be related to the characteristics of our tertiary care centre that offers specific haematology care. Therefore, the data presented here might not be representative of the management of such patients in primary or secondary care centres. An exception was, however, observed in patients with hormone-driven polycythaemia, since 53% of them were not seen by a haematologist.

Treatment recommendations of secondary polycythaemia are different from those of PV [2]. We can assume that, during polycythaemia work-up, preventive treatment measures were introduced by applying PV recommendations, which, afterwards, by excluding PV, were stopped. The cut-off level of Hb/Hct to perform phlebotomy in JAK2 unmutated polycythaemia is usually set higher and is strictly intended to treat symptoms of hyperviscosity and is based on clinical well-being, while cytoreductive treatments are usually not indicated. The lack of clear recommendations on the topic is reflected in the variety of management implemented in this cohort. Thus, 18% of patients received phlebotomy at diagnosis and two patients received cytoreductive treatment and, as expected, during follow-up, a twofold drop in treatment was observed. This study describes a real-world approach of a large cohort of patients with unmutated JAK2 polycythaemia but was not designed to make recommendations on the management of these.

Historically, the management of non-PV erythrocytosis has been conflicted by unfounded concerns regarding the risk of thrombosis, arising from a poor characterization of congenital erythrocytosis forms, except for Chuvash polycythaemia, known for its thrombotic tendency [2]. In our cohort, nonfatal thrombotic events (venous or arterial) occurred in 17% of patients during follow-up. The most frequent venous thromboembolic event was pulmonary embolism, followed by deep venous thrombosis, whereas strokes and myocardial infarctions were the predominant arterial events. There were no differences in thrombotic event frequency among the five subgroups (Table 2). However, clinical data of those events, thrombophilia test results and cardiovascular risk factors potentially contributing to thrombotic risk were not available, representing a clear limitation of this study. There was no difference in patients suffering from thrombotic events between the three age groups. Moreover, the venous and arterial thrombotic events were more frequent (21% and 27%, for venous and arterial, respectively) than in a large study of idiopathic polycythaemia patients with a long follow-up, where the prevalence was 4% [26]. We have not observed any thrombotic events in patients with mutations causing congenital polycythaemia identified by NGS, which is in line with the literature [26]. In the general population, the incidence of thromboembolism increases exponentially with age and is rare in young adults, and very rare before the age of 20 [27]. Therefore, we cannot exclude a relationship between JAK2 unmutated polycythaemia, the relatively high prevalence (6%) of venous and arterial thrombotic events in patients aged 0–30 years.

In contrast to what is observed in PV, we did not observe any transformation of polycythaemia into any other disease. The overall mortality rate was around 3% within a median follow-up period of 47 months. We do not have data regarding causes of deaths, and we do not know whether this is different when compared to a matched general population. However, the observed mortality is clearly lower than for PV, reported as 10% during a large observational study [28].

The strength of this study is the enormous number of patients screened, the confirmation of polycythaemia with Hb/Hct results, the availability of a negative JAK2 test in the patients investigated and access to clinical data of all of them, which makes the cohort very consistent. This study has, however, a number of limitations, mainly related to its retrospective nature and low patient number in some subgroups. In the screened population with confirmed polycythaemia by Hb/Hct laboratory values, JAK2 mutation testing was not performed in 74% of all patients, therefore these were excluded from the analysis. We assume that possibly there was another clinical cause found to explain polycythaemia, however we cannot exclude that, if JAK2 had been tested in all of them, the prevalence or the repartition of JAK2 unmutated polycythaemia would have been different. We also cannot rule out that some patients may have been misdiagnosed as having another chronic myeloid neoplasm. Furthermore, the diagnostic procedure, as well as the decision to treat an individual patient, was made at the discretion of a leading physician. Therefore, there are missing data, since not all diagnostic tests were performed, and in some patients the reason to treat was not always comprehensible. Moreover, no data regarding therapy and risk factors of thromboembolic events, such as cardiovascular diseases, laboratory data including presence of thrombophilia, especially antiphospholipid antibodies or classification of events as provoked or unprovoked, was available. Moreover, FBC results were not consistently available at the time of thromboembolic events in all patients. Likewise, although the deaths could be identified, thanks to the reports in the insurance database, their causes were not available.

In conclusion, this study shows that once PV is ruled out as a cause of polycythaemia by genetic testing, the search for causes of unmutated JAK2 polycythaemia is complex and, in one third of the patients, it may not be found. This study also illustrates that for the work-up of patients with polycythaemia, an orderly diagnostic approach is needed with a major awareness of different possible underlying causes. We find that some diagnostic algorithms are helpful in the resulting interpretation of many but not in all cases. We suggest, therefore, particularly in young patients, to consider common causes such as sleep apnoea, which should not be underestimated, and in some cases the investigation of hereditary causes, using new diagnostic methods such as NGS, should be considered.

## Figures and Tables

**Figure 1 jcm-11-03393-f001:**
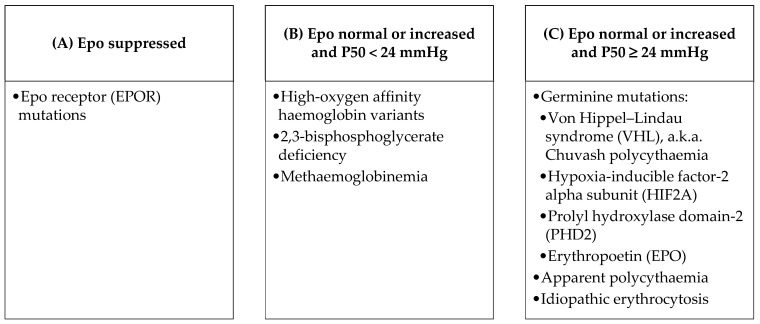
Potential causes of hereditary polycythaemia according to Epo and P50 levels, as reported in the literature.

**Figure 2 jcm-11-03393-f002:**
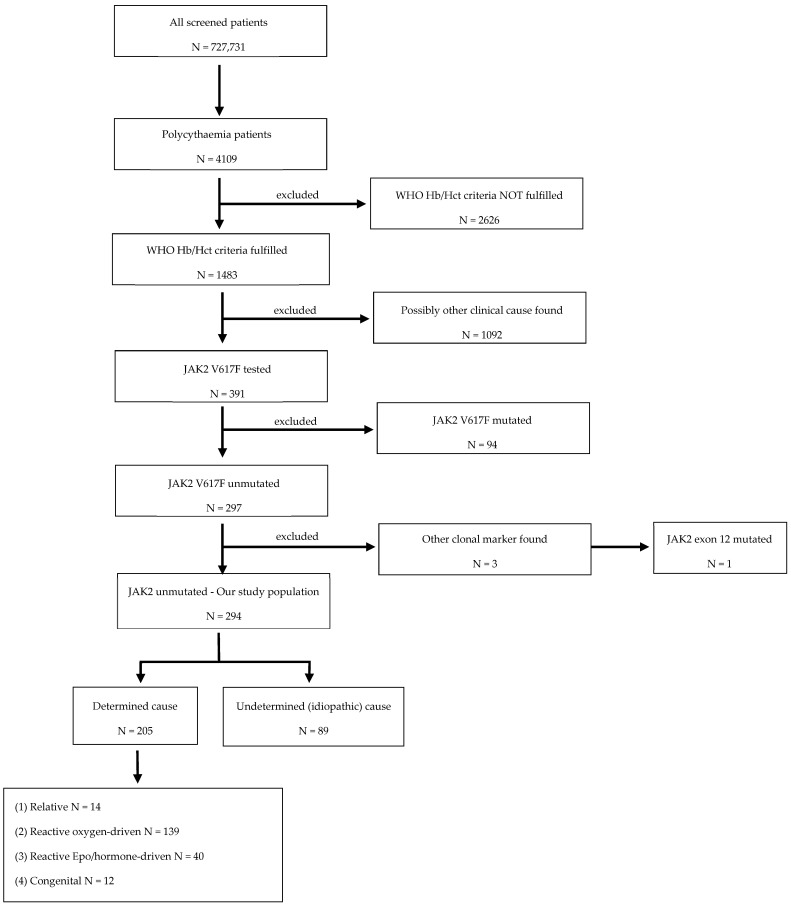
Polycythaemia. Study population, selection criteria for the study period 1 October 2008–31 July 2019.

**Figure 3 jcm-11-03393-f003:**
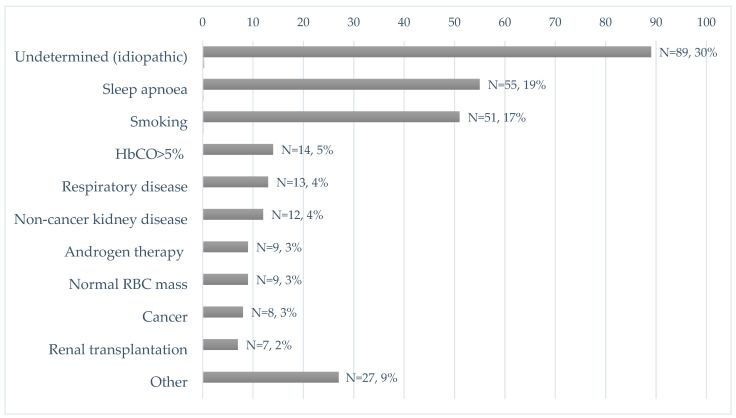
Single causes of JAK2 unmutated polycythaemia (N = 294), according to frequency.

**Figure 4 jcm-11-03393-f004:**
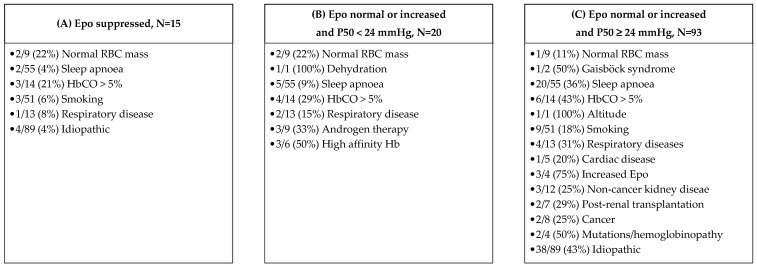
Causes identified in relation to Epo level and P50 levels. Numbers and percentages are reported in relation to the number of patients with the particular cause. Zero entries are omitted. Patients with missing values are not included.

**Figure 5 jcm-11-03393-f005:**
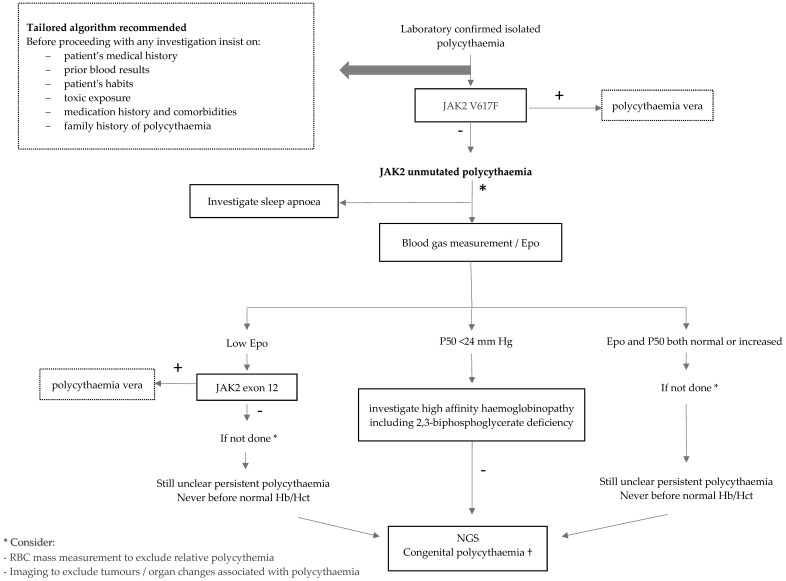
Our proposed diagnostic algorithm of JAK2 unmuted polycythaemia in adults. † Our 13-gene NGS panel includes: EPOR (exons 7,8), VHL (orf including exon 1’), EGLN1, EPAS1, EPO (including several regulatory regions), JAK2 (exons 9–16), BPGM, HBB, HBA1, HBA2, HIF3A, OS9, and SH2B3 (somatic).

**Table 1 jcm-11-03393-t001:** Characterisation of underlying pathophysiological mechanism of polycythaemia: causes, demographics and laboratory values of JAK2 unmutated polycythaemia patients (N = 294).

Type of Polycythaemia		N, All	% Male	Age, Median Years (Range)	Hb, Median g/L (Range)	Hct, Median T/T (Range)	RBC, Median 1012/L (Range)
**All**		**294**	**82**	**46 (15, 89)**	**172 (157, 224)**	**0.51 (0.45, 0.68)**	**5.67 (4.63, 8.65)**
**1. Relative (group 1)**		**14**	**71**	**55 (22, 70)**	**170 (163, 182)**	**0.5 (0.47, 0.65)**	**5.67 (4.92, 7.49)**
	1.1. Normal RBC mass	9	67	59 (25, 68)	169 (163, 182)	0.5 (0.47, 0.57)	5.69 (4.95, 6.69)
	1.2. Dehydration (clinical diagnosis)	1	100	22	176	0.51	5.79
	1.3. Gaisböck syndrome	2	50	49 (35, 63)	175 (170, 180)	0.52 (0.5, 0.53)	5.23 (4.92, 5.53)
	1.4. Capillary leak syndrome	2	100	54 (37, 70)	170 (170, 170)	0.58 (0.5, 0.65)	6.57 (5.64, 7.49)
**2. Reactive oxygen driven (group 2)**		**139**	**80**	**47 (17, 89)**	**175 (157, 224)**	**0.51 (0.46, 0.68)**	**5.71 (4.72, 8.66)**
	2.1. Sleep apnoea	55	89	49 (20, 89)	176 (161, 199)	0.52 (0.47, 0.59)	5.77 (4.85, 7.06)
	2.2. HbCO >5%	14	64	52 (18, 69)	176 (157, 203)	0.52 (0.46, 0.58)	5.94 (4.72, 6.67)
	2.3. Altitude	1	100	43	179	0.5	6.06
	2.4. Smoking	51	76	45 (18, 71)	173 (158, 198)	0.5 (0.46, 0.58)	5.56 (4.74, 6.89)
	2.5. Respiratory disease	13	69	47 (17, 72)	173 (165, 207)	0.51 (0.47, 0.68)	5.84 (5.21, 8.66)
	2.6. Cardiac disease	5	80	40 (21, 50)	184 (163, 224)	0.53 (0.49, 0.67)	5.43 (5.09, 6.6)
**3. Reactive Epo/hormonal driven (group 3)**		**40**	**90**	**55 (23, 82)**	**171 (164, 199)**	**0.51 (0.46, 0.6)**	**5.58 (4.86, 6.68)**
	3.1. Increased Epo	4	75	68 (31, 76)	174 (166, 199)	0.53 (0.5, 0.6)	5.55 (4.86, 6.03)
	3.2. Noncancer kidney disease	12	100	52 (23, 76)	173 (166, 191)	0.52 (0.48, 0.56)	5.52 (5.08, 6.28)
	3.3. Post-renal transplantation	7	100	59 (32, 69)	167 (164, 186)	0.51 (0.47, 0.53)	5.56 (5.15, 6.3)
	3.4. Androgen therapy	9	100	50 (25, 69)	173 (166, 195)	0.52 (0.47, 0.58)	5.7 (5.32, 6.67)
	3.5. Cancer (renal, adrenal, seminoma, lung, brain)	8	62	68 (32, 82)	179 (164, 191)	0.49 (0.46, 0.59)	5.62 (5.18, 6.68)
**4. Congenital (group 4)**		**12**	**75**	**33 (16, 71)**	**172 (161, 184)**	**0.51 (0.47, 0.54)**	**5.45 (4.63, 7.33)**
	4.1. Mutations/hemoglobinopathy	4	100	29 (22, 34)	172 (162, 182)	0.52 (0.48, 0.54)	5.90 (5.48, 7.33)
	4.2. Down’s syndrome	2	50	37 (16, 57)	167 (161, 173)	0.49 (0.47, 0.5)	5.0 (4.84, 5.07)
	4.3. High affinity Hb (increased P50)	6	67	45 (28, 71)	172 (161, 184)	0.51 (0.48, 0.53)	5.28 (4.63, 5.94)
**5. Undetermined (idiopathic) (group 5)**		**89**	**85**	**43 (15, 77)**	**171 (159, 193)**	**0.5 (0.45, 0.56)**	**5.65 (4.87, 6.75)**

**Table 2 jcm-11-03393-t002:** Thrombotic events and therapy during follow-up in JAK2 unmutated polycythaemia patients, N = 294.

Type of Polycythaemia		N	Age, Median Years (Range)	Thrombotic Events			Therapy Received			Therapy Ongoing		
				All, N	Venous, N	Arterial, N	P, N	ASS, N	CRT, N	P, N	ASS, N	CRT, N
**All**		**294**	**46 (15, 89)**	**65**	**33**	**32**	**53**	**19**	**2**	**13**	**12**	**1**
**1. Relative**		**14**	**55 (22, 70)**	**6**	**2**	**4**	**5**	**2**	**0**	**0**	**1**	**0**
	1.1. Normal RBC mass	9	59 (25, 68)	0	0	0	5	2	0	0	1	0
	1.2. Dehydration (clinical diagnosis)	1	22	3	0	3	0	0	0	0	0	0
	1.3. Gaisböck syndrome	2	49 (35, 63)	3	2	2	0	0	0	0	0	0
	1.4. Capillary leak syndrome	2	54 (37, 70)	0	0	0	0	0	0	0	0	0
**2. Reactive oxygen driven**		**139**	**47 (17, 89)**	**32**	**18**	**14**	**20**	**7**	**0**	**4**	**3**	**0**
	2.1. Sleep apnoea	55	49 (20, 89)	15	6	9	11	5	0	2	2	0
	2.2. HbCO >5%	14	52 (18, 69)	0	0	0	2	0	0	1	0	0
	2.3. Altitude	1	43	0	0	0	0	0	0	0	0	0
	2.4. Smoking	51	45 (18, 71)	10	6	4	5	2	0	1	1	0
	2.5. Respiratory disease	13	47 (17, 72)	5	5	0	2	0	0	0	0	0
	2.6. Cardiac disease	5	40 (21, 50)	2	1	1	0	0	0	0	0	0
**3. Reactive Epo/hormonal driven**		**40**	**55 (23, 82)**	**7**	**5**	**2**	**6**	**1**	**0**	**2**	**1**	**0**
	3.1. Increased Epo	4	68 (31, 76)	1	1	0	1	0	0	0	0	0
	3.2. Noncancer kidney disease	12	52 (23, 76)	1	0	1	2	1	0	2	1	0
	3.3. Post-renal transplantation	7	59 (32, 69)	1	1	0	1	0	0	0	0	0
	3.4. Androgen therapy	9	50 (25, 69)	2	1	1	2	0	0	0	0	0
	3.5. Cancer (renal, adrenal, seminoma, lung, brain)	8	68 (32, 82)	2	2	0	0	0	0	0	0	0
**4. Congenital**		**12**	**33 (16, 71)**	**3**	**1**	**2**	**3**	**1**	**1**	**3**	**0**	**0**
	4.1. Mutations/hemoglobinopathy	4	29 (22, 34)	0	0	0	3	0	0	3	0	0
	4.2. Down’s syndrome	2	37 (16, 57)	1	1	0	0	0	0	0	0	0
	4.3. High affinity Hb (increased P50)	6	45 (28, 71)	2	0	2	0	1	1	0	0	0
**5. Undetermined (idiopathic)**		**89**	**43 (15, 77)**	**17**	**7**	**10**	**19**	**8**	**1**	**4**	**7**	**1**

Abbreviations: P—phlebotomy (=venesection), ASS—aspirin, CRT—cytoreductive therapy (hydroxycarbamid, Interferon alfa).

## Data Availability

For this study, there are no publicly archived datasets analysed or generated during the study.

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
