# Peer review of "JAK2 Unmutated Polycythaemia—Real-World Data of 10 Years from a Tertiary Reference Hospital"

_jcm, 2022, doi:10.3390/jcm11123393_

Round 1

Reviewer 1 Report

·       This retrospective analysis underscores the complexity of the search for causes of polycythemia/erythrocytosis in JAK2 unmutated cases. A particular issue is in adolescents and young adults ( AYA-15-30 yrs of age) in whom  congenital causes are  likely. As authors point out, JAK2 mutated cases constitute a very small faction overall and unlikely in the  AYA population. After the initial screening with blood gases and EPO and exclusion of secondary causes- a detailed NGS panel may be the second step as illustrated by the finding of mutations in  the mechanosensitive calcium ion channel gene- PIEZO1 (  doi: 10.1080/08880018.2019.1637984;  doi: 10.1182/blood.2020008424; DOI: 10.1182/bloodadvances.2020003028; doi.org/10.3390/genes12081150).  A brief review on the association of  PIEZO1 mutations and erythrocytosis should be included in the discussion section.

Specific points

Abstract ( Include WHO criteria for polycythemia)

Under causes of polycythemia- clarify “nuclear measurement”- is it by radio labeling studies for red cell mass?

Suggest listing the genes included in the NGS panel as a foot note  to figure 5.

Does the panel include globin genes as standard HPLC evaluation by HPLC may miss some  high oxygen affinity globin mutations?

Consider reviewing causes of polycythemia detected in AYA population ( 15-30 yrs of age, as some may be misdiagnosed to have MDS/MPN without a broader NGS panel.

The risk for thromboembolic disease is well emphasized - please clarify the risk with congenital causes vs others.

Author Response

Reviewer 1, comments:

  1. “A brief review on the association of PIEZO1 mutations and erythrocytosis should be included in the discussion section.”

RESPONSE: We would like to thank reviewer 1 for this comment. The association of PIEZO1 mutations and erythrocytosis has been included in the discussion section starting with line 416: “The list of 13 genes included in the panel, as described elsewhere (16), can be found in the footnote to Figure 5. It includes haemoglobin variants with increased oxygen affinity, due to mutations in the α- or β-globin genes (HBB, HBA2, HBA1). However, it does not include mutations in PIEZO1 gene. Mutations in this gene have been known to cause erythrocyte volume disorders especially hereditary xerocytosis (HX)/dehydrated stomatocytosis (DHS), which are believed to cause mild-to-moderate haemolytic anaemia. Though, a study from 2019 with a large series of patients with HX and PIEZO1 mutations reported that 68% did not show any sign of anaemia and seven patients had polycythaemia (23). Also, a study from 2021 found HX, with well compensated haemolysis, in 4% of investigated idiopathic erythrocytosis  (24). The authors concluded about the need to include the search for a mutation in the PIEZO1 gene in children and young adults with idiopathic erythrocytosis and more generally when erythrocytosis appears before the age of 50 years, especially when associated with iron overload, splenomegaly, elevated mean corpuscular haemoglobin concentration, increased reticulocytes, haemolysis, or decreased P50 (after excluding a variant of Hb with high oxygen affinity). Only recently, our laboratory included investigations on PIEZO1 gene mutations.”

  1. “Abstract - Include WHO criteria for polycythaemia.”

RESPONSE: We would like to thank reviewer 1 for this suggestion. The WHO criteria for polycythaemia were now included into the abstract (lines 18-19).

  1. Under causes of polycythaemia- clarify “nuclear measurement”- is it by radio labelling studies for red cell mass?

RESPONSE: We would like to thank reviewer 1 for pointing this out. Now, an explanation was added in line 130. “RCM and plasma volume were obtained using chromium labelled RBC and albumin respectively.”

  1. “Suggest listing the genes included in the NGS panel as a foot note to figure 5.”

RESPONSE: As suggested by reviewer 1, a detailed description of our “in-house” NGS polycythaemia panel has been included as footnote to figure 5. “Our 13-gene NGS panel includes: EPOR (exons 7,8), VHL (orf including exon 1’), EGLN1, EPAS1, EPO (including several regulatory regions), JAK2 (exons 9-16), BPGM, HBB, HBA1, HBA2, HIF3A, OS9, and SH2B3 (somatic).”

  1. Does the panel include globin genes as standard HPLC evaluation by HPLC may miss some high oxygen affinity globin mutations?

RESPONSE: Thank you for this comment. We agree with the reviewer regarding the limitations of using HPLC in such cases, so we do not use HPLC systematically, and our approach is to test directly with our available NGS panel. As described above (description of the NGS panel in footnote to figure 5), our NGS panel includes mutations in α- or β-globin genes (HBB, HBA1, HBA2). A brief comment referring to the limitations of HPLC investigating high affinity haemoglobins has been added now in line 425: “As a result of limited diagnostic potential using High Performance Liquid Chromatography (HPLC) (23) as it may not detect high oxygen affinity hemoglobinopathies, our NGS panel includes mutations in α- or β-globin genes (HBB, HBA1, HBA2) and the use of HPLC is not part of our algorithm.”

  1. Consider reviewing causes of polycythaemia detected in AYA population (15-30 yrs of age, as some may be misdiagnosed to have MDS/MPN without a broader NGS panel.

RESPONSE: We appreciate this important comment. An explanation was added in line 246. “In 17 young patients with persistent isolated polycythaemia at the end of follow-up the cause remained unexplained”. We also have added line 545 in the discussion in the part of study limitations: “We also cannot ruled-out that some patients may have been misdiagnosed as having another chronic myeloid neoplasms.”

  1. The risk for thromboembolic disease is well emphasized - please clarify the risk with congenital causes vs others.

RESPONSE: We would like to thank reviewer 1 for this comment. We have now included a short paragraph on this subject at the beginning of the paragraph in which we discuss our results on thrombosis (line 513): ”Historically, the management of non-PV erythrocytosis has been conflicted by un-founded concerns regarding the risk of thrombosis, arising from a poor characterization of congenital erythrocytosis forms, except for Chuvash polycythaemia, known for its thrombotic tendency (2).”

Reviewer 2 Report

Exhaustive study investigating a difficult clinical problem.  The introduction, the methods and the discussion are adequate.  The results are not unexpected, but a periodic visit of the topic is necessary, as there advances and changes not only in the diagnostic criteria of the hematolymphoid neoplasm, but also in the identification of alternative etiologies of polycythemia.

If the authors wish, it can be improved by extra tests investigating the calreticulin and MPL status of the idiopathic cases, and discussing the role of next generation sequencing in these cases and which would be the genes to be included in these panels.  The fact that the cases were selected from the records of a tertiary care institution in an affluent country could also have an impact on the results, as one could assume that at least a few cases are referrals that have already selected for difficulty.

In conclusion, while the study does not add much in terms of new information, it can be very useful to practicing physicians managing patients with polycythemia.

Author Response

Reviewer 2, comments:

  1. If the authors wish, it can be improved by extra tests investigating the calreticulin and MPL status of the idiopathic cases, and discussing the role of next generation sequencing in these cases and which would be the genes to be included in these panels. 

RESPONSE: We want to thank for this important comment. We would like to clarify that we have reviewed in all patients all available molecular investigations, including CAL and MPL. In fact, we were forced to exclude 3 patients from our cohort who had other molecular markers. These patients are mentioned in Figure 2. Furthermore, the Ethic Comity Approval for this specific study did not give permission to conduct any further research on biological material, we could only collect data retrospectively.

  1. The fact that the cases were selected from the records of a tertiary care institution in an affluent country could also have an impact on the results, as one could assume that at least a few cases are referrals that have already selected for difficulty.

RESPONSE: Thank you for this comment, we totally agree with this aspect. This is included in the discussion section, starting in line 490 “This high percentage may be related to the characteristics of patients themselves, in whom JAK2-testing was considered necessary, but it may also be related to the characteristics of our tertiary care centre that offers specific haematology care. Therefore, the data presented here might not be representative of the management of such patients in primary or secondary care centres”. (Now highlighted in yellow in the manuscript)

  1. In conclusion, while the study does not add much in terms of new information, it can be very useful to practicing physicians managing patients with polycythaemia.

RESPONSE: Thank you for this comment.

Reviewer 3 Report

The study “JAK2 unmutated polycythaemia – real-world data of 10 years from a tertiary reference hospital” by Katarzyna Aleksandra Jalowiec et al. is an intersting one. The authors have tried to explore the underlying reasons for JAK2 unmutated polycythaemia compared to polycythaemia vera (PV) and have proposed a work-up plan for better diagnosis and management of this clinical issue. The study has employed strategies to screen a large number of previously registered patients over a 10 years period and evaluated their clinical records to identify the genuine JAK2 unmutated polycythaemia cases. The detailed analysis of this cohort of patients to identify the underlying causes of this clinical condition has produced some very intriguing outcomes. Although, there are some limitations to this study such as, its retrospective nature, insignificant number of patients in some subgroups of the disease, and exclusion of a large number of patients due to lack of Jak2 mutation analysis etc., the authors have reported a number of positive outcomes from their analysis. Most importantly, sleep apnoea has been identified as a major cause of polycythaemia in a significant proportion of patients, which is interesting. Despite the detailed analysis the authors have undertaken, they found that unknown reasons still constitute the major proportion of polycythaemia cases. This underscores for further research and careful management of patients.

However, the authors need to improve the ‘introduction’ as it lacks the basic information about polycythaemia in general for readers who are not familiar with it. A long-term follow up than the 47 months reported in the study, would have been better in monitoring the disease status, the treatment strategies adopted, the side effects generated, which will help in planning better management strategies for patients.

The study is methodologically strong and the conclusions drawn are appropriate. Overall, the study has widened our understanding about JAK2 unmutated polycythaemia, and has made some useful suggestions such as use of NGS for better diagnosis of mutations to identify polycythaemia patients.

Author Response

Reviewer 3, comments:

  1. However, the authors need to improve the ‘introduction’ as it lacks the basic information about polycythaemia in general for readers who are not familiar with it.

RESPONSE: We would like to thank the reviewer for pointing this out. A short introduction has been now included into the abstract and in the introduction (lines 14 and 37 respectively): “Polycythaemia is defined by an increase in haemoglobin (Hb) concentration, haematocrit (Hct) or red blood cell (RBC) count above the reference range adjusted to age, sex and living altitude.”

  1. A long-term follow up than the 47 months reported in the study, would have been better in monitoring the disease status, the treatment strategies adopted, the side effects generated, which will help in planning better management strategies for patients.

RESPONSE: We agree with the reviewer that a longer follow-up would have provided greater insight on the subject, however, for this study the follow-up period was defined by the time between the first available and last available blood results. For this reason, the follow-up period cannot be extended. According to these criteria the observed median follow-up time was 47 months, ranging from 0.4 to 258 months (21.5 years).

  1. The study is methodologically strong and the conclusions drawn are appropriate. Overall, the study has widened our understanding about JAK2 unmutated polycythaemia, and has made some useful suggestions such as use of NGS for better diagnosis of mutations to identify polycythaemia patients.

RESPONSE: We want to thank the reviewer for this positive comment.